# A Smart Algorithm for Personalizing the Workstation in the Assembly Process

**Maja Turk** [ID]**, Miha Pipan, Marko Simic \*** [ID] **and Niko Herakovic**

Faculty of Mechanical Engineering, University of Ljubljana, 1000 Ljubljana, Slovenia;
maja.turk@fs.uni-lj.si (M.T.); miha.pipan@fs.uni-lj.si (M.P.); niko.herakovic@fs.uni-lj.si (N.H.)
**\*** Correspondence: marko.simic@fs.uni-lj.si; Tel.: +386-1-4771-727

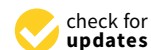

**Featured Application: A self-configurable assembly workstation with integrated I4.0 technologies and controlled by smart algorithm provides an ergonomic workplace for each worker and prevents errors during assembly.**

**Abstract:** Due to increasing competition in the global market and to meet the need for rapid changes in product variability, it is necessary to introduce self-configurable and smart solutions within the entire process chain, including manual assembly to ensure the more efficient and ergonomic performance of the manual assembly process. This paper presents a smart assembly system including newly developed smart manual assembly workstation controlled by a smart algorithm. The smart assembly workstation is self-configurable according to the anthropometry of the individual worker, the complexity of the assembly process, the product characteristics, and the product structure. The results obtained by a case study show that is possible to organize manual assembly process with rapid adaptation of the smart assembly system to new products and workers characteristics, to achieve ergonomic working conditions through Digital Human Modelling (DHM), to minimize assembly time, and to prevent error during the assembly process. The proposed system supports the manual assembly process redesign to ensure a better working environment and aims to have an important value for applying the smart algorithms to manual assembly workstations in human-centered manufacturing systems.

**Keywords:** manual assembly process; smart algorithm; smart manual assembly system; self-configurable workstation; workplace evaluation

## 1. Introduction

Manufacturing is a process of converting raw materials into finished goods through a production process involving machines, tools and labor (human worker) [1]. With the I4.0 paradigm, the "smart" has been considered as a core characteristic of the future manufacturing system [2]. Smart manufacturing is an emerging form of production integrating manufacturing assets of today and tomorrow with sensors, computing platforms, communication technology, control, simulation, data-intensive modelling and predictive engineering [3,4] to achieve greater efficiency, competency, and competitiveness of industrial processes [5].

The manufacturing industry has gone through many changes, especially in the way of the liberation of human manual labor, [2] but manual work is, and in future will be, an important component of production in any industry, because it is directly related to the productivity of the system. The abilities, skills, productivity and performance of the workers have great importance for the increase of production [6–8], so there is a need to develop a multi-skilled workforce capable of performing multiple tasks [9] or to develop a worker-centered system with digital instructions [10,11]

or a virtual training system [12] to replace extensive training and lack of experience. On-site training and support are essential to help workers master the required skills, improve worker productivity and ensure product quality [13]. In addition to skills, workers' performance is closely linked to their working environment. The design and planning of manual assembly is a complex and time-consuming process, as many areas of study need to be involved simultaneously, such as anthropometry, vision, acoustics, time analysis, ergonomic assessment, cognitive requirements, technical and economic factors, etc. [14–16]. Assembly workers are exposed to a significant physical workload. The repetitiveness of manual tasks is an important risk factor for upper limbs Work-related Musculoskeletal Disorders (WMSD) and back health problems [17]. Improved workplace design enables the assembly operators to work safely, thus reducing hazardous and strenuous reaches and preventing potentially serious injuries. Workplace design and ergonomics can greatly influence assembly efficiency and increase worker productivity, ensure worker safety, physical and mental well-being and job satisfaction during manual assembly operations [6,16,18]. The use of computer-aided techniques, in particular the use of Digital Human Modelling (DHM), is very useful for overcoming some of the difficulties encountered in workplace design [19–21]. It facilitates a complete data-based ergonomic analysis and provides a strong scientific validity of the solutions proposed or implemented [1].

There is a lot of research in the literature that focuses on different approaches for designing assembly systems, especially manual assembly systems. In Cohen et al. [22] it is stated, that as a consequence Industry 4.0 could strongly impact on the actual assembly paradigms, even in the cases in which the human factor is prevailing, as manual assembly. The authors have investigated how the transformations to I4.0 principles are expected to occur. Moreover, for our study and for the implementation of I4.0 principles, the Operator Support Systems (OSS) and Self-Adapting Smart Assembly System (SASS) are crucial. The authors stated that the main principles behind the design of an efficient assembly system 4.0 are: connectivity (connect and collect data whenever and wherever you can), information (from the collected data create usable information), knowledge (create knowledge from Operator Support System (OSS)), and "smart" (make the assembly system a Self-Adapting Smart System (SASS)). "Smart" means that the assembly system shall adapt through the use of smart actuators as a consequence of the generated information and according to algorithms. Bortolini et al. [23] propose an original framework which investigates the impact of the principles of Industry 4.0 on assembly system design named "Assembly system 4.0". The main features of purposed systems are late customization of assembled products, assembly control system, aided assembly, intelligent storage management, product and process traceability and self-configured workstation layout. For our research, the feature "self-configured workstation layout" is relevant. The proposed self-configured workstation layout autonomously adjusts the rack, shelf and workbench dimensions considering the assembly product and the assigned worker. Self-configured workstation layout aims to optimize the assembly activities and minimize picking and fastening time thus ensuring ergonomic working conditions. Favi et al. [24] proposed Design for Assembly (DfA) approach, called 4M, which takes into account all the most important aspects involved in the manual assembly: Method (assembly issues related to the assembly procedure), Machine (assembly issues related to the workstation layout), Man (assembly issues related to the worker) and Material (assembly issues related to the product and components design). The aim of their study is to provide a means for the concurrent improvement of the product design and assembly line, the workstation ergonomics, and assembly tasks. In our study, we use a holistic approach to design an assembly system (smart workstation, smart algorithm, implementation of tool and technologies, inclusion of ergonomics), like Favi et al. in [24] for product design. Bertram et al. [8] give an overview of existing solutions and prototypes in the field of assistive systems for manual working stations in research and practice and discuss their specific focus. All systems focus on an information assistant that provides the right information for the right situation. Only a few solutions and projects are presented here in the paper: Augmented Workplace from motionEAP project (development of an augmented workplace), ProMiMo (implementation of user-centered assistance for manual assembly at a

workbench), Manual Working Station of SmartFactory KL (development of smart workstation equipped with an assistive system for sequential assembly process), Operator support system TNO (Bosch, development of operator support system for assembly, focusing on information assistance for the worker), Active Assist Bosch Rexroth (system serving as a configurable and open web platform. Features of this software include a context-sensitive information provision and a standardized interface for additional system components (e.g., pick-to-light, projector, touchscreen, Radio-frequency identification (RFID) reader). Shikdar et al. [18] developed a fully adjustable, ergonomically designed assembly workstation with special features such as a motorized table with upward, downward and angular movements, an ergonomic chair with the adjustable seat pan, arm and back supports, and a mechanism for bins and tool adjustments. They experimented on traditional and the smart assembly workstations to boost operators' performance and reduce occupational health and safety problems in the workplace. In Gjeldum et al. [25], the authors present the advantages and disadvantages of implementation of I4.0 technologies (RFID, custom made visual guidance on touch liquid-crystal display (LCD), push buttons, etc.) in iterative assembly line balancing process of a gearbox assembly line. Due to complex mechanical parts in gearboxes, the assembly is performed manually. Assembly line balancing process was carried out through the "Manual approach" and "I4.0 approach". The analyses showed the unsurpassed advantage of "I4.0 approach" in comparison to "Manual approach" in term of increase of information flow speed, reduction of paper sheet consumption, reduction of necessary working space for administrative activities, and simplification of data post-processing. Zamfirescu et al. [26] focused on the visualization and digital guidance of the manual assembly process. They claim that with increasing complexity and variability of products, the main difficulty for the operator is to follow the right assembly procedure to manufacture the desired product correctly. To support the procedure, there are many possible alternatives for setting up a smart working environment (i.e., pick/put-by-light/voice/vision systems) to guide the human operator during the assembly process. There are also alternative recommendations, like augmented reality (AR), virtual reality (VR) instructions [10,13,27,28] and projector-based digital instructions [27]. In smart assembly processes there are more studies reporting the use of AR application to assist workers through assembly process in comparison to VR applications. One of the possible reason is described in ElMaraghy et al. [29]. Authors explained that augmented reality technologies allow the user to see the real environment with virtual objects superimposed upon the real world. AR is potentially cheaper and more realistic tool for assembly design than VR. AR systems can utilize established techniques in VR and evaluate the assembly more realistically with a combination of virtual and real objects in the actual workplace. Lai et al. [10] propose a smart instruction system with the support of AR and deep learning-based tool detection, which is intended to improve worker performance through assistive instructions. To evaluate the integrated AR system, they compared the paper manual provided by the manufacturer of the computer numerical control (CNC) carving machine and the AR instruction. The result of usage of smart instruction system showed reduction of completion time and the number of assembly errors compared to the paper manual. Wang [28] emphasizes AR as the information-level real-time visualization method (IRV), which transforms real-time constraints into computer-generated real-time graphics, called information level real-time AR instruction (IRAI). In a case study, they analyzed and compared the performance of participants under real-time AR instructions with different information levels. The research shows that a high cognitive level of IRAI can make individuals have better performance in work efficiency and cognitive efficiency. To my best knowledge, this is the first user study to evaluate the results of real-time visual information design, so there are many potential research areas in the future. The evaluation of AR visualization assistive systems in reported papers showed common results that untrained users can assemble products faster and with a lower error rate, which is the goal in the industry and the proposed research.

Therefore, our proposal and the aims of the research are to: (i) develop a smart algorithm that controls newly designed smart workstation with smart technologies and digital instructions for assembly tasks, which is flexible and fully adjustable and eliminates the limitations of the

traditional workstation, (ii) evaluate the functionalities of the smart workstation with the performance of the operator, such as time and number of errors compared to the traditional workstation, and (iii) ergonomically evaluate reaching motion with the DHM approach. The advantage of the proposed system is the self-configuration of the entire assembly workstation for the needs of an individual worker, according to gender and body height (constitution), to ensure the ergonomic performance of the assembly process. Moreover, the usage of the system is simple enough, so that thorough training in technology and operation is not necessary. This means that the assembly process can be carried out by an untrained worker without delays and errors.

From the literature it can be understood that the need for human labor is great and that only with a properly designed work environment it is possible to perform jobs efficiently to consequently increase productivity in the industry. However, current researches are mainly focused on the partial solutions of manual assembly workstations, like the development of AR applications (visualization), implementation of individual technologies (VR, pick-by-light, RFID traceability, DHM simulations) and not on the overall configuration of the manual assembly workstation. The implementation of smart technologies at a manual assembly workstation does not yet make the workplace suitable for the highly productive work of an individual worker. When designing a workplace, the constitution and abilities of the individual worker, the properties of the products and the complexity of the assembly process must be taken into account.

Therefore, this is a challenge we addressed and also the main contribution of our study, i.e., a holistic approach to the design of the working environment, both in terms of designing a manual assembly system (workstation and algorithm) and by considering the individual worker (algorithm, ergonomics). So, only by implementing a smart algorithm that controls smart technologies and adapts the workplace for an individual worker, we can achieve an increase in productivity, ergonomic suitability of the designed workstation, errors prevention during the assembly process (even for untrained workers) and consequently, a higher added value per employee.

This paper is structured as follows: Section 2 focuses on the presentation of the smart assembly workstation, smart algorithm and case study description. The Section 2.1 includes presentation of the smart manual assembly workstation and its main implemented technologies and its operational functions. The next subsection presents the smart algorithm in details and Section 2.3 includes workstations', products', and workers' characteristics involved in the case study. Section 3 presents the results and discussions about time and error analysis and ergonomic evaluation of the workstations. Section 4 further summarizes the final findings of the study.

## 2. Materials and Methods

The smart system consists of a smart algorithm that controls a smart manual assembly workstation with implemented smart technologies and tools. The system guides the individual worker through the entire assembly process with digital instructions and appropriate visualization. As stated in Qu et al. [30] the functions of SMSs (smart manufacturing systems) are as follows: Self-sensing function (capturing the data and the critical information from the environment), Self-organizing function (the capacity of solving the emergent requirements), Self-deciding function (data-driven decision making process in manufacturing), and Self-adaptive function. Self-adaptive function operates the behavior of SMSs' elements, which are based on the real-time sensing data and information. Through the different algorithm and rules, self-adaptive behavior of SMSs is considered as continuous learning or life-long learning. As a primary intelligence level, adaptivity implies the capacity of acting on rules. "If-Then-Else", as is the primary rule that we follow during the process of developing our smart system with smart algorithm.

Section 2 is divided into three parts. The first part contains the description of the smart workstation, the second part contains presentation of the smart algorithm and its functioning, and the third part presents the overview of the case study, which includes a description of the smart and traditional workstation and the characteristics of the products and workers who performed the experiment.

*2.1. Smart Assembly Workstation*

When choosing the technology for the construction of a manual assembly workstation (Figure 1), the main guide was the low price and availability of the products. We used two Raspberry Pi 3B+, manufactured by Sony UK Technology Centre, Pencoed, UK, (RPi) servers with Raspbian GNU/Linux 9.6, manufactured by Raspberry Pi Foundation, Cambridge, UK for the manual assembly workstation control system. Two RPI controllers were used because we wanted to build a modular manual assembly workstation. The code is written in the Python programming language. The desired current hardware states on manual assembly workstation are written in the Microsoft SQL Express database, and there the hardware states are recorded by the program LASIM product management (LPM) at each step of the assembly. The Raspberry Pi controllers are connected to the base via the network and send a motion signal to the appropriate hardware when the hardware state changes.

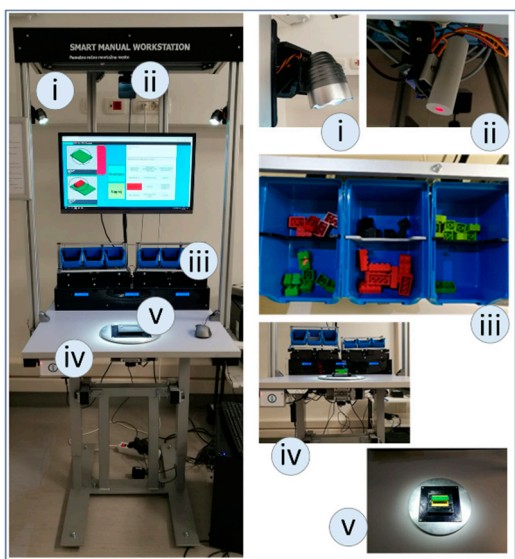

**Figure 1.** Smart manual assembly workstation with implemented tools.

We have illuminated the work surface with additional adjustable lights (Figure 1), (i) due to the possibility of adjusting the intensity and direction of the light beam. The lights are controlled by a Pulse-width modulation (PWM) signal, which enables the control of light intensity. The lights have their own 12 V power supply. The lights are mounted on a two-axis stand with radio control (RC) servomotors. These are connected to Raspberry Pi via the RC servo motor/I2C interface card. The laser pointer (pick-by-light (Figure 1)), (ii), which points to the area in the buffer, is also mounted on the same stand and connected in the same way. The storage of parts contains two separate containers (left and right: LZ, DZ) each with 6 sections (LZ1-6, DZ1-6) driven by a linear servo motor (Figure 1), (iii). They are also connected to the Raspberry Pi controller. We used adjustable table (Figure 1), (iv) to raise the work surface to the appropriate height for each worker. The switches for lifting the table were connected to Raspberry Pi via a relay board. The table had its own power supply by plugging it into a socket. The rotatable table (Figure 1), (v) has a stepper motor and a 24 V power supply. The stepper motor is controlled by the Arduino controller via the stepper motor interface.

For safety reasons, movement is limited to slow speeds and low power motors are used to avoid injury. RC Servomotors generate typical noise, which could be reduced by using high-quality motors. Replacing these motors with better motors would also increase the accuracy and repeatability of laser and light positioning.

*2.2. Smart Algorithm*

According to the literature review, manual assembly is developing mainly in the direction of AR [10,13,26], but visualization itself through AR does not ensure higher productivity and ergonomic suitability of the workplace. To achieve these results and reduce the number of errors in assembly operations and shorten the time of work operations due to human error, it is necessary to design a self-configurable smart manual assembly workstation with smart technologies for error prevention, visualization and an implemented smart algorithm that connects the technologies and adapts the workplace depending on the selected influencing parameters. The smart algorithm includes three main groups of influential parameters: human, assembly process and product, as shown in Figure 2.

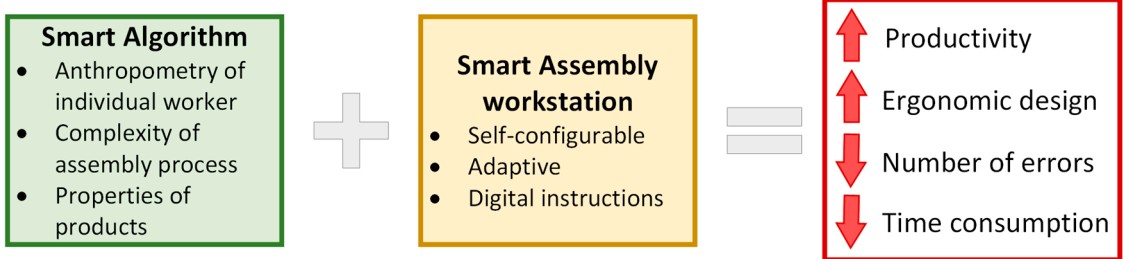

**Figure 2.** Smart algorithm and smart assembly workstation for improved productivity of the assembly process.

The algorithm is the core of a smart manual assembly workstation. It is based on the dependencies within the influential parameters, which include the individual worker, the type or complexity of the assembly and the product/part properties.

Figure 3 shows the dependencies between the influential parameters of the smart algorithm and their influence on the control of the smart manual assembly workstation. Using a smart algorithm, we control the height of the workbench, lighting and grab containers with parts and rotation of the assembly nest. The anthropometric characteristics takes directly into account the body height (with standard deviation), and indirectly the hand range, elbow height, eye height, shoulder-grip length, and additionally the gender of the worker [31–33]. We wanted to use directly as little data as possible, due to data storage and the law on personal data protection (GDPR, [34]), so we wrote down all indirectly used anthropometric features in such a form that they functionally depend only on gender and body height. The anthropometric characteristics and gender influence the working height as well as the distance and inclination of the grab containers [32,33] in which the components are stored. The complexity of the assembly was divided into normal work, work with heavy objects and precise work—work with smaller parts. The complexity of the assembly affects the work, i.e., the work surface rises according to the original calculation from human dimensions when the precise assembly operation is performed and lowers when working with heavier and larger objects. If the assembly complexity is normal, the height of the workbench remains at the previously set height [33].

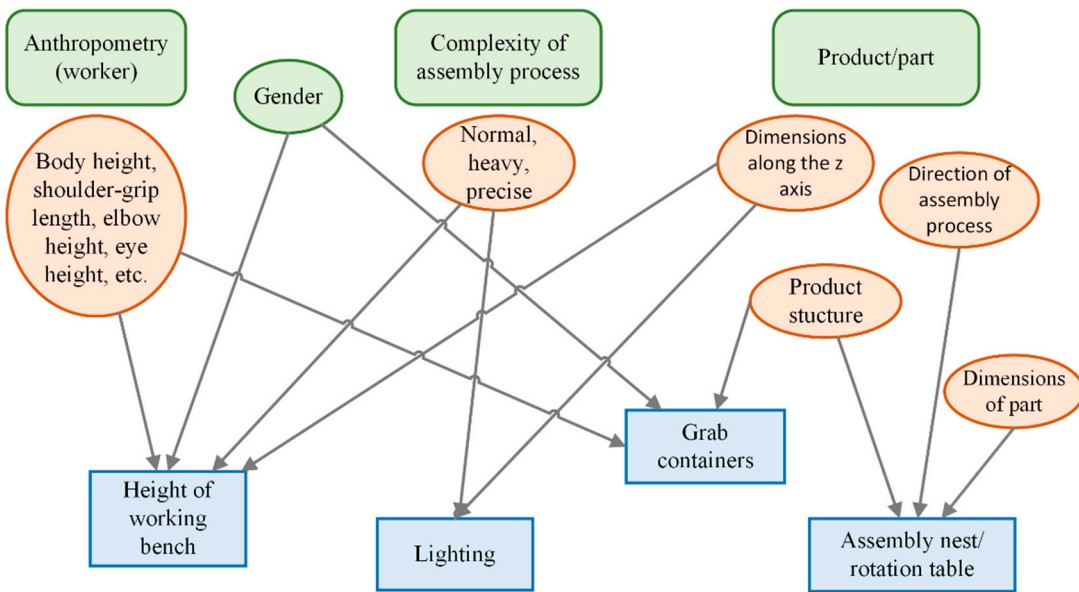

**Figure 3.** Dependence between the influential parameters of the smart algorithm and their impact on the control of the smart manual assembly workstation.

The complexity of the assembly also affects the illuminance, because the assembly nest is more strongly illuminated when the precise assembly is performed and less strongly when large objects are assembled [35,36]. The third set of influential parameters refer to the product or part; these include the dimensions of the base part, the dimensions of the product along the *z*-axis (height of the product), the direction of assembly and the product structure tree. The height of the product influences the illumination direction (light beam), as the height of the product changes the position of the assembly process along the *z*-axis so that the focus of the illumination must be adjusted according to the "new" assembly position. The height of the product also affects the height of the working bench. If the semi-finished product reaches a height of 100 mm during the assembly process, the work surface must be lowered by 100 mm so that we maintain the recommended working height. The assembly direction affects the rotation of the assembly nest, as we want the worker to always add new parts from the same direction. All influential parameters are connected by the product structure, assembly instructions and the sequence of assembly operations of individual parts.

The structure of the product is displayed to the worker in the form of digital instructions and is shown directly on the screen of the smart manual assembly workstation. The worker communicates and confirms the performed operations by clicking the button (B_next) until the assembly process of the product is completed, which is determined by the end button (B_finish). Figure 4 shows the block diagram of the algorithm. The worker logs on with the RFID chip to the smart assembly workstation where the RFID reader is mounted. The chip contains the worker's ID number, which is connected to the SQLite database, from where the worker's anthropometric data is stored. These data are used to determine the "working height 1" and the distance and inclination of the grab containers based on empirical equations and recommendations [31–33]. Then follows the assembly process for the selected product according to the work plan. Each product has labels that are written into the product structure at each step (Figure 5).

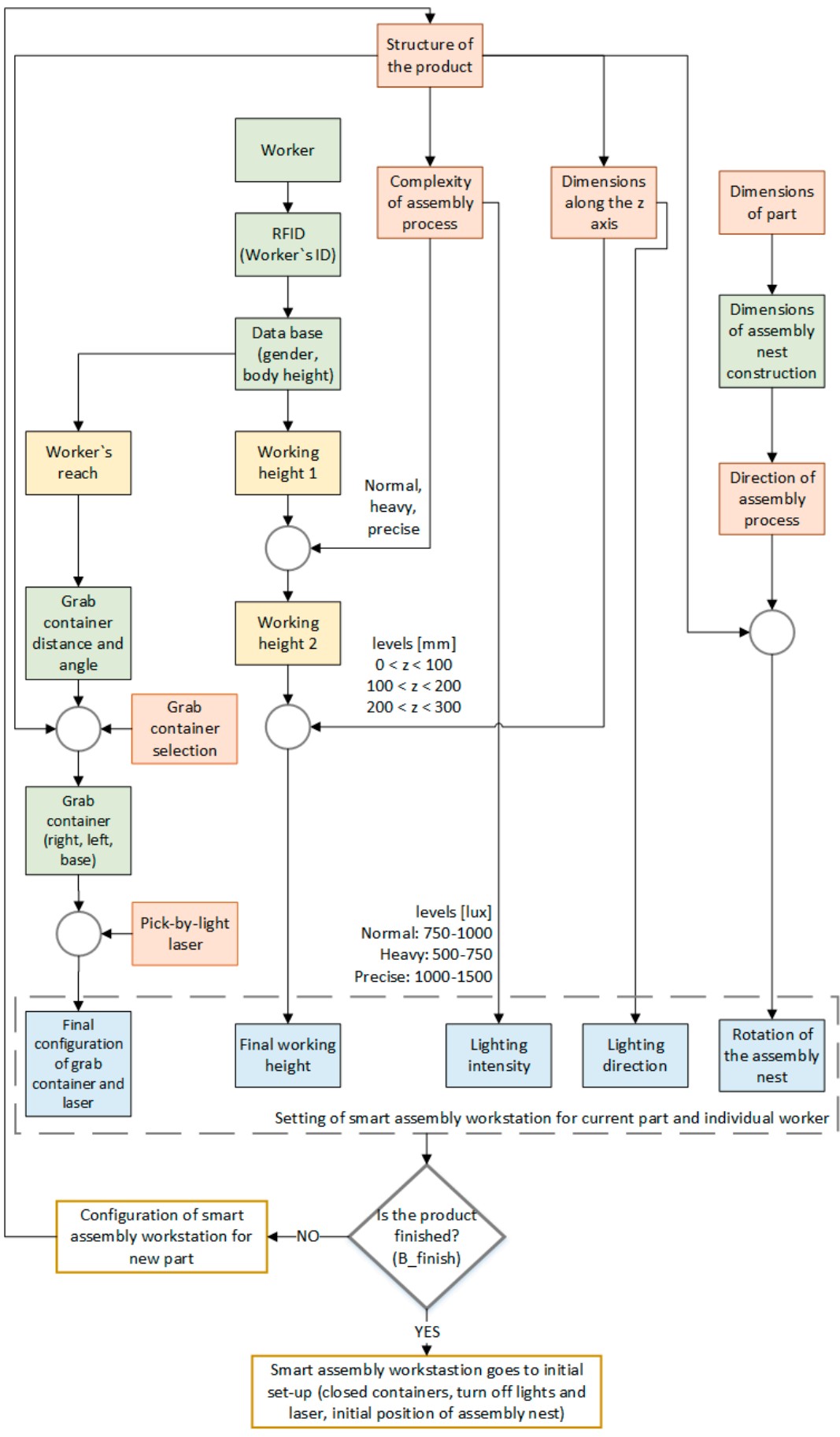

**Figure 4.** Block diagram of the smart algorithm.

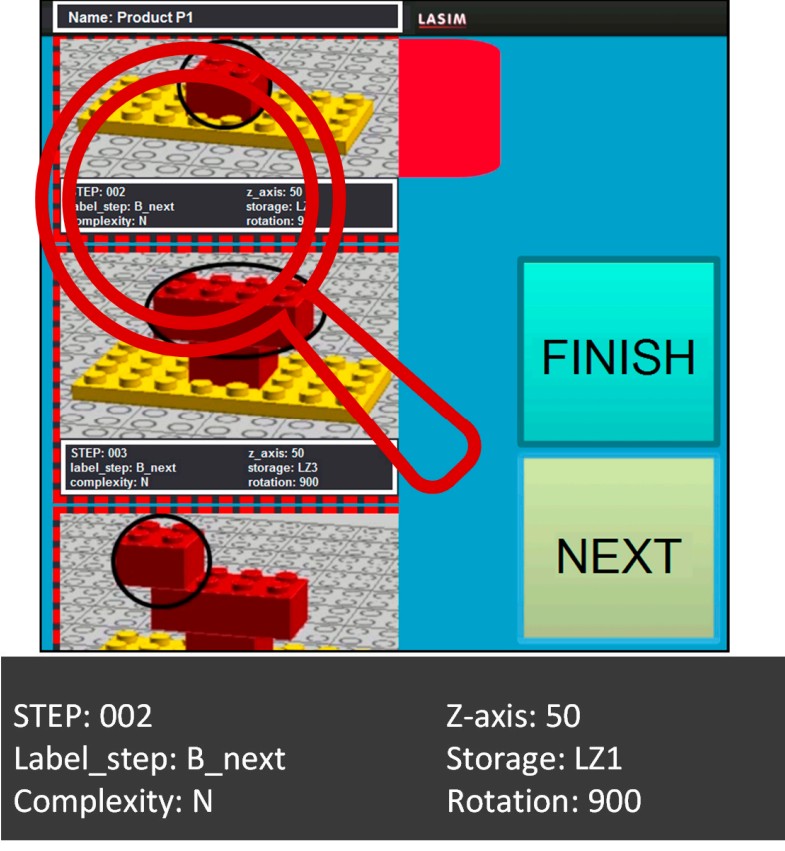

**Figure 5.** Labels that are written for each step of the product structure.

The algorithm reads them and writes their values to the SQLite database. Accordingly, the operation and configuration of the smart assembly workstation are determined. The labels written at each step are (1) assembly step, intermediate steps (B_next), final step (B_finish); (2) assembly complexity, "N" = "Normal", "P" = "Precise", "H" = "Heavy"; (3) dimensions along the z-axis: "50" mm = when the assembly operation for this step starts, the height of the semi-finished product is between 0 mm and 100 mm, "150" mm = when the assembly operation for this step starts, the height of the semi-finished product is between 100 mm and 200 mm, "250" mm = when the assembly operation for this step starts, the height of the semi-finished product is between 200 mm and 300 mm, etc., (4) the grab containers in which the current part is stored: BZ, LZ1-6, DZ1-6; "BZ" means the base part, "LZ" means the left container, "DZ" means the right container and 1–6 are the sections inside the container, (5) rotary table for the assembly nest: the label indicates the angle of rotation; "900" means the angle of rotation 90 °, "1800" means 180 °, etc. Continuing the determination of the final height of the workbench, as described and shown in the block diagram, also consists of reading labels from the database on the complexity of assembly and the dimensions of the product along the z-axis. To determine the configuration of the grab containers and the laser beam (Pick by Light) for the current step of the assembly operation, the data of the location of the current part (BZ, LZ1-6, DZ1-6) is obtained from the database (distance and inclination are determined in advance). Based on the obtained data, the grab container is opened at a certain distance, tilted by a certain angle and the laser beam is directed to the section of the grab container where the part to be currently assembled according to where the product structure/assembly step is located. The illumination (intensity and direction of the light beam) is adjusted according to the influential parameters read from the database. To determine the angle of rotation of the assembly nest, the information on the assembly direction is read from the database (e.g., 900). The assembly nest is already pre-constructed according to the dimensions of the base part.

*2.3. Overview of the Case Study*

The research aims to present the functionality of a smart assembly workstation controlled by a smart algorithm to analyze assembly process times, ergonomic suitability, and the number of errors.

2.3.1. Traditional vs. Smart Manual Assembly Workstation

The main advantage of a smart manual assembly workstation is that it not only advances in the development of the visualization but also takes over the entire assembly process, i.e., it takes into account the individual worker, prevents errors with appropriate instructions and smart technologies and takes care of ergonomic suitability of the workplace. On this basis, the worker can carry out error-free assembly of the product without additional training in process and technology. Figure 6 shows the methodology of the case study, which is divided into a description and functioning of a traditional and smart manual assembly workstation, as well as a description of experimental work involving workers, the product and a DHM simulation to determine ergonomic suitability. The study provides results on the time required to perform the assembly process, the number of errors at a smart and traditional workplace during the assembling the product and the assessment of the ergonomic design.

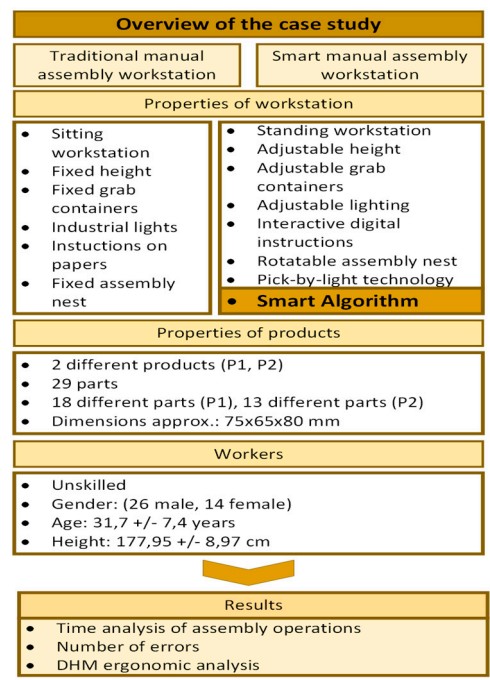

**Figure 6.** Overview of the case study.

The functionalities of traditional workstations in the case study are (i) seated workstation, (ii) height-adjustable classic chair, (iii) workbench at a fixed height, (iv) fixed position and angle of the grab containers, (v) industrial lighting, but unfortunately too weak with an illuminance of 166 lux for detailed work with small parts (the recommendation for this work is 1000–1500 lux [36]), (vi) instructions on the papers so that the workers have to turn the pages and follow the assembly sequence all by themselves, and (vii) fixed assembly nest (Figure 7a).

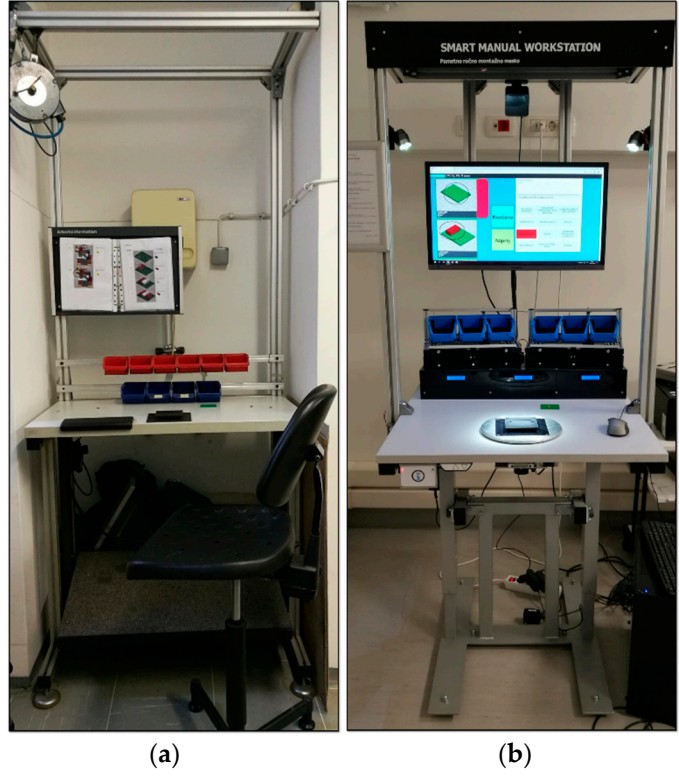

(**a**)          (**b**)

**Figure 7.** (**a**) Traditional assembly workstation; (**b**) smart assembly workstation.

The functionalities of our smart assembly workstation are beside the intelligent algorithm: (i) optional sitting/standing workstation, (ii) height-adjustable workbench (set at the height recommended by [33,37]) for each worker), (iii) distance and angle adjustable grab containers. They move within the reach of the worker if they contain a part by product structure for the assembly operation, or they move out of reach if they do not contain the part required by the assembly structure. Six grab containers, which are attached to the same rails are adjusted simultaneously. To avoid the mistakes of finding the right part in one of these six grab containers, the smart manual workstation also features (iv) Pick-by-Light technology. This technology guides the worker through the structure of the product and displays the grab container with the relevant part that the worker needs to use during assembly. Other smart technologies and tools implemented include (v) adjustable lights that independently focus on the assembly nest, (vi) interactive digital instructions, named LPM software that guides the worker through the assembly process/product structure; it is only the worker's job to confirm the completed assembly task by clicking the "Next" button, and (vii) rotatable assembly nest that rotates according to the optimal direction of the assembly operations. We have also implemented AR and VR on a smart workstation, but these technologies are not relevant to the experiment in this study (Figure 7b).

### 2.3.2. Characteristics of Products and Workers

Two different products were selected for the experiment (Figure 8). Both have 29 parts. Product P1 has 18 different parts and products P2 has 13; the parts are Lego blocks configured as shown in Figure 8. We have selected products that have a different degree of complexity (different number of parts) and variability but are representative and easy to assemble. The assembly of the products was carried out by untrained employees (regardless of gender, age or anthropometric characteristics) both at the traditional manual assembly workstation and at the smart one. At each assembly workstation (Traditional and Smart) 40 experiments were carried out, 20 for each product. The experiments were performed by PhD students and laboratory staff. They carried out the experiment for the first

time and were recruited voluntarily. The experiment was performed by 40 subjects ($N = 40$, gender: 26 male, 14 female). Their mean (SD) anthropometric data were: age 31.7 (7.4) years; height 1780 (90 mm). Everyone was instructed to perform the experiment with a speed that was feasible for the entire work shift, i.e., the entire working life of a person. The experiments were recorded for further processing.

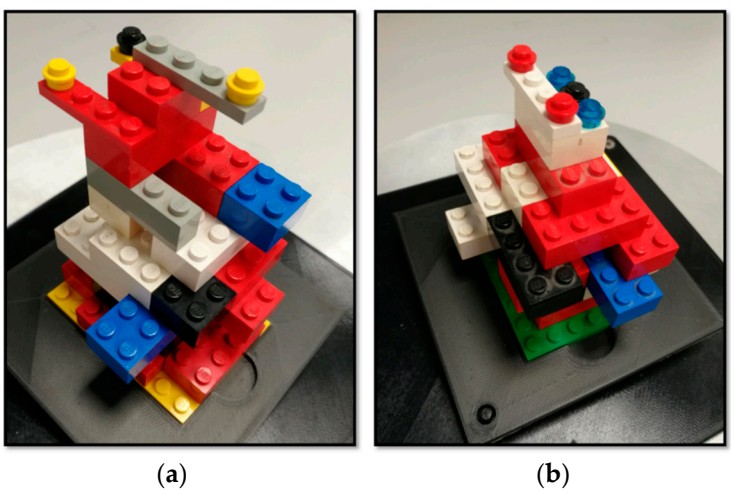

(**a**) 　　　　　　　　　　　　(**b**)

**Figure 8.** (**a**) Product P1; (**b**) product P2.

To evaluate the ergonomics, we used the software package Siemens Jack DHM, into which we imported computer-aided design (CAD) models of both workstations. We used a male avatar with the body height of 1780 mm (as the average height of experiment participants) as a virtual human and performed a motion analysis/reach analysis for the entire assembly process of the product P1. We chose to analyses product P1 because it is more complex than product P2.

## 3. Results and Discussion

The Results and Discussion section is divided into two main subsections: results of smart algorithm and results of the experiment based on time analysis, error detection and evaluation of ergonomics.

### 3.1. Smart Algorithm

Figure 9 shows the block diagram of a smart algorithm for one step of the assembly process for the actual worker who participated in the research. The values calculated by the smart algorithm for the individual configuration of the controlled components of the smart assembly workstation using empirically determined equations (as in the recommendations [31–33]) and the dependencies between the influential parameters are shown in bold. The block "Structure of the product" shows that we used step 002 as an example. Along with the block diagram there are labels (marked in bold) read by the smart algorithm from the SQLite database: (1) worker: Female, 1710 mm tall; (2) grab container and pick by light laser: LZ1; (3) complexity of assembly: N (Normal); (4) dimension along the *z*-axis: 50 (at this step, the height of the subassembly is 15 mm; (5) rotation of assembly nest: 900; (6) button: button is defined in the decision block, the answer is "NO", so the label is not B_finish, therefore the label is B_next. Previously, we took into account the size of the base part and designed an assembly nesting with appropriate dimensions. The results show the configuration of the smart manual assembly workstation: (1) grab container: left, distance: 500 mm from the shoulder, angle: 20°, section: 1, (2) height of the working bench: 905 mm, (3) lighting: intensity: 830 lux, direction: center of the assembly nest; (4) rotation of the assembly nest: angle: 90°, and the following step, step 003, where the algorithm is executed according to a block diagram from the beginning.

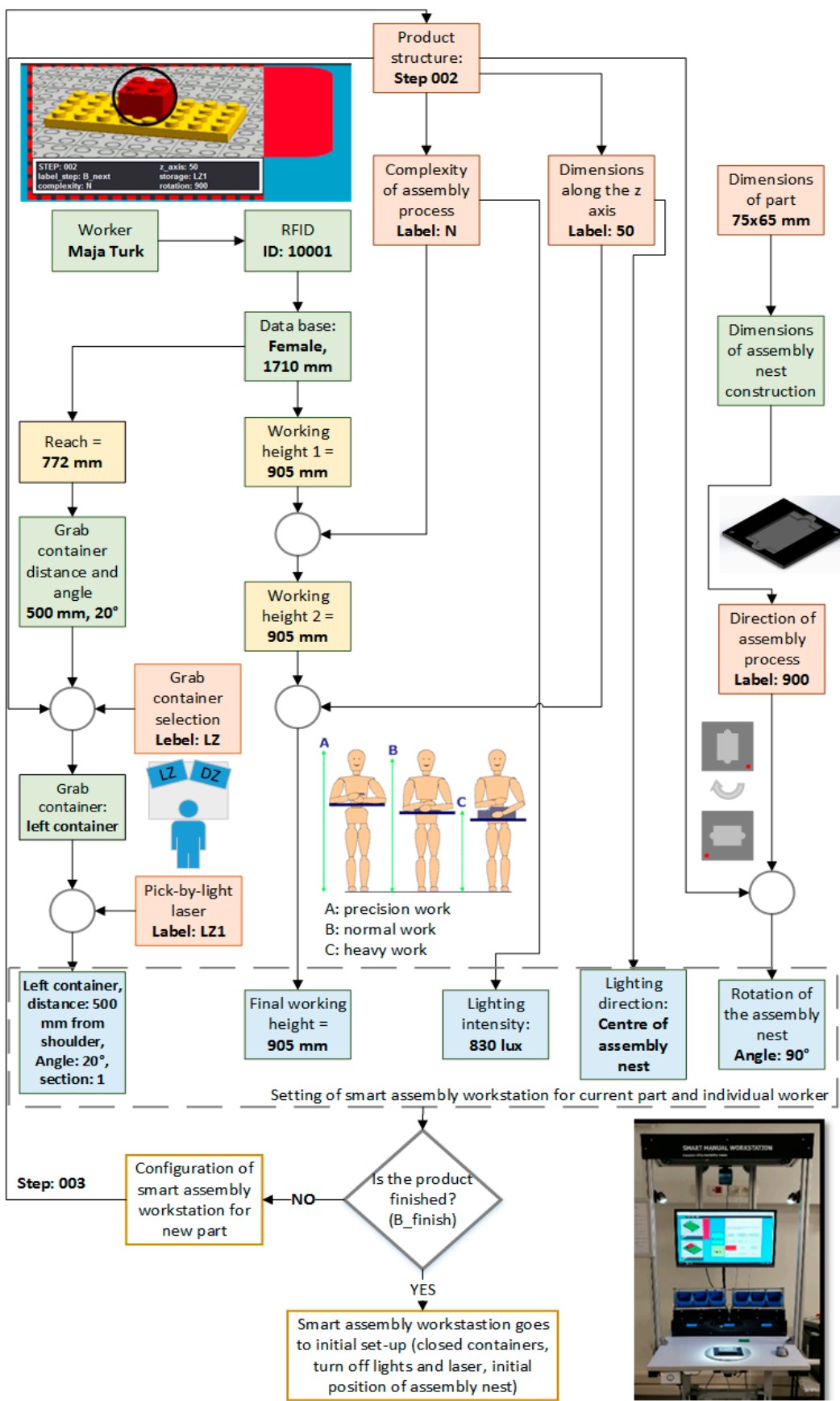

**Figure 9.** Block diagram of a smart algorithm for step number 002.

*3.2. Laboratory Experiment*

This section is divided into three parts: time analysis of the assembly process conducted at traditional and smart assembly workstations, comparison of error and error prevention techniques and ergonomic evaluation of workstation design.

### 3.2.1. Time Analysis

The results of the comparison of assembly times for the assembly of products P1 and P2 on traditional and smart manual workstations are shown in Table 1. At the smart manual workstation, all workers completed the assembly faster than at the traditional workplace. Product P1 is assembled at the traditional manual workstation in an average of 216.8 s, which is 24% longer than at the smart workstation (175.3 s). For product P2, workers spend an average of 187.4 s on the traditional workstation, which is 14% longer than the results on the smart workstation (159.7 s). The time analysis also shows that P1 is more complex, and this complexity means that more time is spent on the assembly process. It should be emphasized here that the reduced time is only the result of the implementation of the new assembly system and was not the primary goal of the research, so we do not want workers to have more stringent time norms. This result is an indication that the proposed assembly system makes the workplace with the worker more efficient. After analyzing the videos of the experiment and the workers' user experience, we found two significant time wastages on the traditional manual assembly workstation. First, when workers search for parts in the grab containers, and second when workers do not follow the instructions or get lost reading the assembly sequence from the paper instructions.

**Table 1.** Measured times of experiments at traditional and smart manual assembly workstation.

| Product Name | Traditional Manual Assembly Workstation | Smart Manual Assembly Workstation |
| :---: | :---: | :---: |
| | Final time (SD) (s) | Final time (SD) (s) |
| P1 | 216.8 (59.5) | 175.3 (53.5) |
| P2 | 187.4 (39.2) | 159.7 (51.6) |

Disadvantages of a smart assembly workstation that result from reviewing videos and interviews with the workers are:

- The existing LPM software has a small delay between the display of the current step and the next step, which in the assembly process is shown as the worker waiting for further instructions. This leads to time wastage and lowers added value and productivity.
- When the worker assembles the product for the first time, he or she receives a lot of information (lights on, pick-by-light technology, LPM, rotation of the assembly nest, movement of grab containers), which causes initial confusion. However, this is only a transitional state, which on average lasts just for two assembly steps.
- The LPM software displays the current and the next assembly operations in parallel, which has confused some workers to follow the next instructions without clicking the "Next" button after completing the current step.
- The laser beam emitted by the pick by light technology should be more intense, as it is not sufficiently visible on the blue bottom of the grab containers.

We have considered all constructive criticisms and limitations related to the current smart assembly workstation, which provide a good basis for improvements and the definition of development guidelines in future work.

### 3.2.2. Number of Errors

The results shown in Table 2 represent the number of errors made by workers during the assembly of products P1 and P2 at traditional and smart manual assembly workstations. It has been shown

that with the implementation of a smart algorithm that controls smart technologies and tools at the workstation and guides the worker through the assembly process, the number of errors is reduced for more than 72%. Errors at the traditional manual assembly workstation are usually caused by workers who have assembled the wrong parts or the right parts in the wrong place. All workers have discovered and successfully corrected the errors themselves. However, errors on the smart manual assembly workstation occurred when the right component was assembled in the wrong product location. All workers discovered the errors themselves and corrected them successfully. At the smart manual assembly workstation, the errors where a worker reaches into the grab containers for parts were eliminated. On the contrary, the errors that occur during the assembly process itself are still present. In further work, the segment of error elimination will be addressed by upgrading with the AR technology.

**Table 2.** The number of errors occurred at traditional and smart manual assembly workstation.

| Product Name | Traditional Manual Assembly Workstation | Smart Manual Assembly Workstation |
|:---:|:---:|:---:|
| P1 | 11 | 3 |
| P2 | 11 | 4 |
| | 22 | 7 |

### 3.2.3. Ergonomic Evaluation

The ergonomic analysis of the reaching was carried out in the DHM environment of the Siemens Jack. We imported CAD models of manual assembly workstations into the virtual environment, added a predefined avatar and designed the entire assembly process as in a real environment. Figure 10 shows a graphical representation of the distribution of forward reaches per work cycle at a traditional assembly workstation and a CAD model of a traditional workstation with an avatar in a virtual environment. The traditional manual assembly workstation is designed according to the standards so that we can compare the distribution of the forward reaches with the smart manual assembly workstation. The results show that the worker reaches 12 times too far, 16 times within acceptable range and only once in close range. At close range, the entire product is assembled, so there is no storage space. However, it is ergonomically and medically less appropriate for a worker to reach too far in 41% of the 29 assembly operations (according to ergonomic regulations [38]).

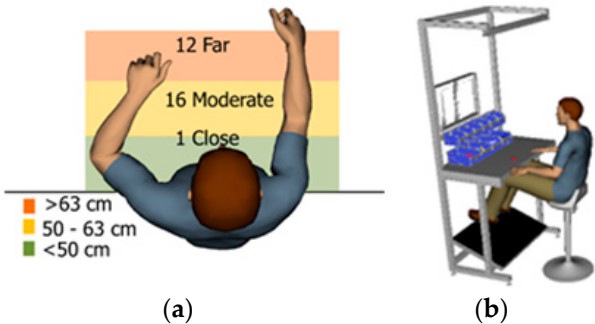

(**a**)          (**b**)

**Figure 10.** (**a**) Distribution of forward reaches per work cycle on traditional assembly workstation; (**b**) computer-aided design (CAD) model of a traditional workstation with an avatar in a virtual environment.

In the case of the smart manual assembly workstation, we already considered ergonomics when designing the workstation and positioned the grab containers and considered their functionalities more appropriately from an ergonomic point of view using a smart algorithm and empirical equations based on the gender and height of the worker. Figure 11 shows that during the assembly of the same products as at the traditional assembly workstation, the worker never enters the danger zone (far reach). By implementing the adjustable grab containers in this way on the smart manual assembly workstation, we have reduced reaches into the danger zone from 12 to 0 compared to the traditional manual

assembly workstation. The initial position of the grab containers was identical for the traditional and smart assembly workstation. The difference is in the adjustable grab containers, which can change their position to the worker's moderate zone during the assembly process if necessary. In this case, the position of the container is programmed to change its position according to the reach of the worker's hand and the position of the part. With a smart manual assembly workstation, the worker mainly reaches the acceptable zone, which does not affect the deterioration of the worker's health in the long term.

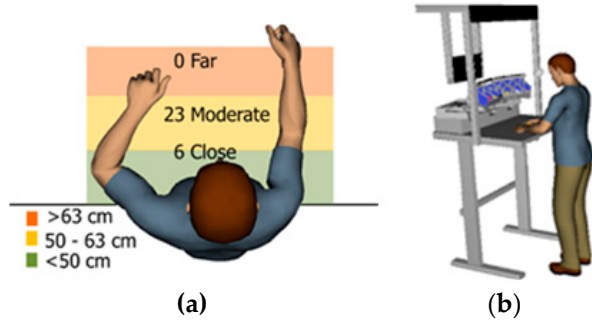

**(a)**          **(b)**

**Figure 11.** (**a**) Distribution of forward reaches per work cycle on smart assembly workstation; (**b**) CAD model of a smart workstation with an avatar in a virtual environment.

## 4. Conclusions

Due to the high involvement of people in the assembly process and the movement towards striving for higher productivity, the need has arisen to adapt the workplace to the workers. In this study, we presented the development of a smart algorithm that takes into account various influential parameters, such as the anthropometry of the individual worker, the properties of the product and the complexity of the assembly process. It controls a smart manual assembly workstation developed in the laboratory and implemented with smart technologies and tools. The main functions of the workstation, which have a great influence on the efficiency of the assembly process, are the height of the workbench, the direction and intensity of the lighting, the rotation of the assembly nest and the distance and inclination of the grab containers, which are controlled by labels. The workstation is also equipped with digital instructions, which are displayed via LPM software and pick-by-light technology, showing the correct section of the grab container where the current part is stored. The newly developed smart assembly workstation has been compared to a traditional fixed workstation in terms of its operational capabilities.

The time evaluation, error analysis and an evaluation of the ergonomic suitability were carried out. The experimental measurements were carried out with untrained and unskilled workers, while the ergonomic evaluation was done in a Siemens Jack virtual environment. The results of the time analysis showed that the main drawbacks of a smart manual assembly workstation are the instruction program, as it is slower than the workers' skills. Without considering this disadvantage, the assembly times of the products P1 and P2 on a smart manual assembly workstation are shorter than on a traditional workstation. The error analysis has shown that the use of smart technologies and tools reduces the number of errors for more than 72%. However, the results showed that we still have not eliminated the errors that occur at the assembly nest when the worker places the right part in the wrong place. An ergonomic analysis of the assembly process of the P1 product has shown that the placement of the grab container at a traditional workstation forces the worker to reach for the parts in the hazardous zone 12 times; on the contrary, at a smart assembly workstation, we have eliminated the worker reaches to the hazardous zone.

The main contribution of the study is presented by all three analyses which have shown that the design or redesign of assembly workstation is recommendable for the health (ergonomic analysis) as well as for the productivity (time, errors) of the worker and thus for productivity improvements of the

companies. Another contribution of the smart workstation design controlled by a smart algorithm and appropriate digital instructions is that we can save time and reduce the cost for in-depth training and further learning of workers since research has shown that the assembly of simple products can be carried out with less errors even by untrained workers in comparison to a traditional workstation.

For the further work, we will: (i) take into account the workers' comments and recommendations that we received during the measurement to improve the smart assembly, (ii) improve the LPM program, and (iii) eliminate the possibility of errors occurring and recurring. The research of smart algorithm also goes towards the implementation of human databases for different nationalities (workers' constitutions) and legalizations for lighting for other countries. Future work will also focus on upgrading the smart algorithm with machine learning methods and visualization using AR technology.

**Author Contributions:** Conceptualization, M.T. and N.H.; methodology, M.T.; software, M.T. and M.P.; validation, M.S. and N.H; formal analysis, M.T. and M.S.; investigation, M.T.; resources, M.T., M.P. and M.S.; data curation, M.T. and M.P.; writing—original draft preparation, M.T.; writing—review and editing, M.T., M.S and N.H.; visualization, M.T.; supervision, M.S. and N.H. All authors have read and agreed to the published version of the manuscript.

**Funding:** The work was carried out in the framework of the GOSTOP programme (OP20.00361), which is partially financed by the Republic of Slovenia, Ministry of Education, Science and Sport Republic of Slovenia (Ministrstvo za izobraževanje, znanost in šport) and European Union, European Regional Development Fund. The authors also acknowledge the financial support from the Slovenian Research Agency (research core funding No. P2-0248).

**Conflicts of Interest:** The authors declare no conflict of interest.

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
