# Peer review of "A Smart Algorithm for Personalizing the Workstation in the Assembly Process"

_applsci, doi:10.3390/app10238624_

Round 1

Reviewer 1 Report

The paper topic is suitable for publication consideration in Applied Science journal.

The aim of the paper is to present developed „smart algorithm“ that modify the smart workstation according to the anthropometric features, the complexity of the assembly process, the product characteristics and the product structure.

In the Introduction section (row 74-75), authors said that the smart application of the proposed algorithm provides improvement of product design? How?

In addition, a summary of the sections constituting the paper must be added at the end of Introduction, nevertheless some information can be found on the beginning of succeeding sections.

For anthropometric features, authors use height and gender. As far as I know, anthropometric features can be specified according to gender, but gender is not an anthropometric feature.

Why the proposed algorithm is considered “smart”? This is the algorithm that, according to the given input (workers’ ID and product types and assembly complexity), adapts the smart workstation (height of the working bench, lighting, grab containers and assembly nest/rotation table) according to the data already entered in the system database. Please clarify.

In chapter 2.1. (row 192-194) as well as in chapter 3.1. (row 306) authors mentioned empirical equations and recommendations. What are these equations? Could you list and explain which equations are in question? How you get them? Or, are those taken from the literature that is not cited?

How did you determine the values of the data you enter into the database on which the customization of smart assembly is based? In particular, the location and angle of a grab container and rotation of the rotary table / assembly nest. How did you determine the optimal direction of an assembly operation?

“Case study methodology“ is somehow an inappropriate title for the section 2.2.. Through this section, I cannot see which scientific methods you will use for your case study so the word “methodology” cannot be used properly. The methodology should show which methods, tools and techniques we will use in proving or rejecting certain hypotheses. Perhaps you can use some other title, for example: “Case study information”, or “A short review of data for the case study”…

Please add a new section where you describe the newly created smart workstation in a little more detail. What does it all entail? Which technology you use on it? Why? How the table, shelves, containers, lights, etc. within the workstation are managed, powered?  Are there some drawbacks noted or reported by workers, as safety issues due to moving elements, increased noise, strain on human vision…

Why you use a predefined avatar (male; height:1774 mm) as a virtual human, and not for example human with the mean height (1780 mm)?

In row 308. you mentioned “labels that are colored red” on figure 8? I don’t see the red labels. Is it important to be presented on picture?

In the conclusion (row 437) you said that “…research has shown that the assembly of simple products can be carried out without errors”. How? According to table 2, errors occurs during the assembly of both of your product (P1 that is more complex and P2 that is a less complex product).

In my opinion, the subject of the manuscript is very timely and widely described in scientific journals, which is why the literature review seems to be poor. Addition of certain number of recent literature is obligatory. For example, paper “Utilization of Industry 4.0 Related Equipment in Assembly Line Balancing Procedure“, Processes 2020, 8, 864, presents only one of reported custom made visual guidance solutions incorporating I4.0 related equipment, nevertheless only limited number of these solutions are mentioned to exists, in rows 78-87. Paper „Assembly system configuration through Industry 4.0 principles: The expected change in the actual paradigms“. IFAC-PapersOnLine 2017, 50, revealing consequence during transformation to Industry 4.0, which could strongly impact on the actual assembly paradigms, including in manual assembly. The paper „Information-level real-time AR instruction: a novel dynamic assembly guidance information representation assisting human cognition”, The International Journal of Advanced Manufacturing Technology volume 107, pages 1463–1481(2020), could presents as leading point in future research, as AR is emphasized…

Please check the text by native English speaker. Already in Abstract, there are sentences which are difficult to follow. Minor typing errors are found.

Overall, the paper needs a major revision, explanations and clarifications. The second iteration of review process is obligatory.

Author Response

Dear Reviewer,

Thanks very much for taking your time to review this article. I really appreciate all your comments and suggestions! Please find my itemized responses in attachment Response to Reviewers and my revisions/corrections in the re-submitted files (using “Track Changes” function).

Thanks again!

Reviewer 2 Report

You can find the comments and suggestions in the attachment.

Author Response

(The authors gave the same response as above.)

Round 2

Reviewer 1 Report

The submission is extended and improved in order to clarify all the concerns and doubts of reviewer. Sufficient explanations in letter to reviewer and inserted text paragraphs in manuscript are provided.

Few minor changes has to be done prior to acceptance:

Gender is still visible under Anthropometry, nevertheless it is not a feature of Anthropometry. Some other data can be inserted to maintain ellipse and arrows, but, as both ellipses points to same rectangles, only one ellipse can be used with generalized data types or short list of data types.

The quality of Figure 7 is not adequate. The dpi is low. If some segments should be hidden from wider audience, those should be blurred. Fig 8. have the same issue.

I am not sure that “the number of errors is reduced by more than 300%“ is aproprirate term. We have for example 11 and 3. When reducing from 11 to 3, we reduced the number of errors for approx. 73%. (or “for more than 72%”)

The provided manuscript is convincing enough, and of adequate quality to be published in Applied Science journal, after few minor changes are made.

Author Response

Dear Reviewer,

Thank you for taking the time to review the article for the second time. I really appreciate your final comments! Please find my responses in below and my corrections in the re-submitted file (using “Track Changes” function and additionally the “Comment” function). The new corrections within the round 2 are commented in the manuscript.

Thanks again!
